# A Stability-based Validation Procedure for Differentially Private Machine Learning

**Kamalika Chaudhuri**
Department of Computer Science and Engineering
UC San Diego, La Jolla CA 92093
kamalika@cs.ucsd.edu

**Staal Vinterbo**
Division of Biomedical Informatics
UC San Diego, La Jolla CA 92093
sav@ucsd.edu

## Abstract

Differential privacy is a cryptographically motivated definition of privacy which has gained considerable attention in the algorithms, machine-learning and data-mining communities. While there has been an explosion of work on differentially private machine learning algorithms, a major barrier to achieving end-to-end differential privacy in practical machine learning applications is the lack of an effective procedure for differentially private parameter tuning, or, determining the parameter value, such as a bin size in a histogram, or a regularization parameter, that is suitable for a particular application.

In this paper, we introduce a generic validation procedure for differentially private machine learning algorithms that apply when a certain stability condition holds on the training algorithm and the validation performance metric. The training data size and the privacy budget used for training in our procedure is independent of the number of parameter values searched over. We apply our generic procedure to two fundamental tasks in statistics and machine-learning – training a regularized linear classifier and building a histogram density estimator that result in end-to-end differentially private solutions for these problems.

## 1  Introduction

Privacy-preserving machine learning algorithms are increasingly essential for settings where sensitive and personal data are mined. The emerging standard for privacy-preserving computation for the past few years is differential privacy [7]. Differential privacy is a cryptographically motivated definition, which guarantees privacy by ensuring that the log-likelihood of any outcome does not change by more than $\alpha$ due to the participation of a single individual; an adversary will thus have difficulty inferring the private value of a single individual when $\alpha$ is small. This is achieved by adding random noise to the data or to the result of a function computed on the data. The value $\alpha$ is called the privacy budget, and measures the level of privacy risk allowed. As more noise is needed to achieve lower $\alpha$, the price of higher privacy is reduced utility or accuracy. The past few years have seen an explosion in the literature on differentially private algorithms, and there currently exist differentially private algorithms for many statistical and machine-learning tasks such as classification [4, 15, 23, 10], regression [18], PCA [2, 5, 17, 12], clustering [2], density estimation [28, 19], among others.

Many statistics and machine learning algorithms involve one or more parameters, for example, the regularization parameter $\lambda$ in Support Vector Machines and the number of clusters in $k$-means. Accurately setting these parameters is critical to performance. However there is no good apriori way to set these parameters, and common practice is to run the algorithm for a few different plausible parameter values on a dataset, and then select the output that yields the best performance on held-out validation data. This process is often called *parameter-tuning*, and is an essential component of any practical machine-learning system.

A major barrier to achieving end-to-end differential privacy in practical machine-learning applications is the absence of an effective procedure for differentially private parameter-tuning. Most previous experimental works either assume that a good parameter value is known apriori [15, 5] or use a heuristic to determine a suitable parameter value [19, 28]. Currently, parameter-tuning with differential privacy is done in two ways. The first is to run the training algorithm on the same data multiple times. However re-using the data leads to a degradation in the privacy guarantees, and thus to maintain the privacy budget $\alpha$, for each training, we need to use a privacy budget that shrinks polynomially with the number of parameter values. The second procedure, used by [4], is to divide the training data into disjoint sets and train for each parameter value using a different set. Both solutions are highly sub-optimal, particularly, if a large number of parameter values are involved – the first due to the lower privacy budget, and the second due to less data. Thus the challenge is to design a differentially private validation procedure that uses the data and the privacy budget effectively, but can still do parameter-tuning. This is an important problem, and has been mentioned as an open question by [28] and [4].

In this paper, we show that it is indeed possible to do effective parameter-tuning with differential privacy in a fairly general setting, provided the training algorithm and the performance measure used to evaluate its output on the validation data together obey a certain *stability condition*. We characterize this stability condition by introducing a notion of $(\beta_1, \beta_2, \delta)$-stability; loosely speaking, stability holds if the validation performance measure does not change very much when one person's private value in the training set changes, *when exactly the same random bits are used in the training algorithm in both cases* or, when one person's private value in the validation set changes. The second condition is fairly standard, and our key insight is in characterizing the first condition and showing that it can help in differentially private parameter tuning.

We next design a generic differentially private training and validation procedure that provides end-to-end privacy provided this stability condition holds. The training set size and the privacy budget used by our training algorithms are *independent of $k$*, the number of parameter values, and the accuracy of our validation procedure degrades only logarithmically with $k$.

We apply our generic procedure to two fundamental tasks in machine-learning and statistics – training a linear classifier using regularized convex optimization, and building a histogram density estimator. We prove that existing differentially private algorithms for these problems obey our notion of stability with respect to standard validation performance measures, and we show how to combine them to provide end-to-end differentially private solutions for these tasks. In particular, our application to linear classification is based on existing differentially private procedures for regularized convex optimization due to [4], and our application to histogram density estimation is based on the algorithm variant due to [19].

Finally we provide an experimental evaluation of our procedure for training a logistic regression classifier on real data. In our experiments, even for a moderate value of $k$, our procedure outperformed existing differentially private solutions for parameter tuning, and achieved performance only slightly worse than knowing the best parameter to use ahead of time. We also observed that our procedure, in contrast to the other procedures we tested, improved the correspondence between predicted probabilities and observed outcomes, often referred to as model calibration.

**Related Work.** Differential privacy, proposed by [7], has gained considerable attention in the algorithms, data-mining and machine-learning communities over the past few years as there has been a large explosion of theoretical and experimental work on differentially private algorithms for statistical and machine-learning tasks [10, 2, 15, 19, 27, 28, 3] – see [24] for a recent survey of machine learning methods with a focus on continuous data. In particular, our case study on linear classification is based on existing differentially private procedures for regularized convex optimization, which were proposed by [4], and extended by [23, 18, 15]. There has also been a large body of work on differentially private histogram construction in the statistics, algorithms and database literature [7, 19, 27, 28, 20, 29, 14]. We use the algorithm variant due to [19].

While the problem of differentially private parameter tuning has been mentioned in several works, to the best of our knowledge, an efficient systematic solution has been elusive. Most previous experimental works either assume that a good parameter value is known apriori [15, 5] or use a heuristic to determine a suitable parameter value [19, 28]. [4] use a parameter-tuning procedure where they divide the training data into disjoint sets, and train for a parameter value on each set. [28]

mentions finding a good bin size for a histogram using differentially private validation procedure as an open problem.

Finally, our analysis uses ideas similar to the analysis of the Multiplicative Weights Method for answering a set of linear queries [13].

## 2  Preliminaries

**Privacy Definition and Composition Properties.** We adopt differential privacy as our notion of privacy.

**Definition 1** *A (randomized) algorithm $\mathcal{A}$ whose output lies in a domain $\mathcal{S}$ is said to be $(\alpha, \delta)$-differentially private if for all measurable $S \subseteq \mathcal{S}$, for all datasets $D$ and $D'$ that differ in the value of a single individual, it is the case that:* $\Pr(\mathcal{A}(D) \in S) \leq e^\alpha \Pr(\mathcal{A}(D') \in S) + \delta$. *An algorithm is said to be $\alpha$-differentially private if $\delta = 0$.*

Here $\alpha$ and $\delta$ are privacy parameters where lower $\alpha$ and $\delta$ imply higher privacy. Differential privacy has been shown to have many desirable properties, such as robustness to side information [7] and resistance to composition attacks [11].

An important property of differential privacy is that the privacy guarantees degrade gracefully if the same sensitive data is used in multiple private computations. In particular, if we apply an $\alpha$-differentially private procedure $k$ times on the same data, the result is $k\alpha$-differential private as well as $(\alpha', \delta)$-differentially private for $\alpha' = k\alpha(e^\alpha - 1) + \sqrt{2k \log(1/\delta)}\alpha$ [7, 8]. These privacy composition results are the basis of existing differentially private parameter tuning procedures.

**Training Procedure and Validation Score.** Typical (non-private) machine learning algorithms have one or more undetermined parameters, and standard practice is to run the machine learning algorithm for a number of different parameter values on a training set, and evaluate the outputs on a separate held-out validation dataset. The final output is the one which performs best on the validation data. For example, in linear classification, we train logistic regression or SVM classifiers with several different values of the regularization parameter $\lambda$, and then select the classifier which has the best performance on held-out validation data. Our goal in this paper is to design a differentially private version of this procedure which uses the privacy budget efficiently.

The full validation process thus has two components – a training procedure, and a validation score which evaluates how good the training procedure is.

We assume that training and validation data are drawn from a domain $\mathcal{X}$, and the result of the differentially private training algorithm lies in a domain $\mathcal{C}$. For example, for linear classification, $\mathcal{X}$ is the set of all labelled examples $(x, y)$ where $x \in \mathbf{R}^d$ and $y \in \{-1, 1\}$, and $\mathcal{C}$ is the set of linear classifiers in $d$ dimensions. We use $n$ to denote the size of a training set, $m$ to denote the size of a held-out validation set, and $\Theta$ to denote a set of parameters.

A *differentially private training procedure* is a randomized algorithm, which takes as input a (sensitive) training dataset, a parameter (of the training procedure), and a privacy parameter $\alpha$ and outputs an element of $\mathcal{C}$; the procedure is expected to be $\alpha$-differentially private. For ease of exposition and proof, we represent a differentially private training procedure $\mathcal{T}$ as a tuple $\mathcal{T} = (\mathcal{G}, F)$, where $\mathcal{G}$ is a density over sequences of real numbers, and $F$ is a function, which takes as input a training set, a parameter in the parameter set $\Theta$, a privacy parameter $\alpha$, and a random sequence drawn from $\mathcal{G}$, and outputs an element of $\mathcal{C}$. $F$ is thus a deterministic function, and the randomization in the training procedure is isolated in the draw from $\mathcal{G}$.

Observe that any differentially private algorithm can be represented as such a tuple. For example, given $x_1, \ldots, x_n \in [0, 1]$, an $\alpha$-differentially private approximation to the sample mean $\bar{x}$ is $\bar{x} + \frac{1}{\alpha n}Z$ where $Z$ is drawn from the standard Laplace distribution. We can represent this procedure as a tuple $\mathcal{T} = (\mathcal{G}, F)$ as follows: $\mathcal{G}$ is the standard Laplace density over reals, and for any $\theta$, $F(\{x_1, \ldots, x_n\}, \theta, \alpha, r) = \bar{x} + \frac{r}{\alpha n}$. In general, more complicated procedures will require more involved functions $F$.

A *validation score* is a function $q : \mathcal{C} \times \mathcal{X}^m \to \mathbf{R}$ which takes an object $h$ in $\mathcal{C}$ and a validation dataset $V$, and outputs a score which reflects the quality of $h$ with respect to $V$. For example, a

common validation score used in linear classification is classification accuracy. In (non-private) validation, if $h_i$ is obtained by running the machine learning algorithm with parameter $\theta_i$, then the goal is to output the $i$ (or equivalently the $h_i$) which maximizes $q(h_i, V)$; our goal is to output an $i$ that approximately maximizes $q(h_i, V)$ while still preserving the privacy of $V$ as well as the sensitive training data used in constructing the $h_i$s.

## 3   Stability and Generic Validation Procedure

We now introduce and discuss our notion of stability, and provide a generic validation procedure that uses the privacy budget efficiently when this notion of stability holds.

**Definition 2** (($\beta_1, \beta_2, \delta$)-**Stability**)  *A validation score $q$ is said to be $(\beta_1, \beta_2, \delta)$-stable with respect to a training procedure $\mathcal{T} = (\mathcal{G}, F)$, a privacy parameter $\alpha$, and a parameter set $\Theta$ if the following holds. There exists a set $\Sigma$ such that $\Pr_{R \sim \mathcal{G}}(R \in \Sigma) \geq 1 - \delta$, and whenever $R \in \Sigma$, the following two conditions hold:*

1. **Training Stability:** *For all $\theta \in \Theta$, $V$, and all training sets $T$ and $T'$ that differ in a single entry, $|q(F(T, \theta, \alpha, R), V) - q(F(T', \theta, \alpha, R), V)| \leq \frac{\beta_1}{n}$.*

2. **Validation Stability:** *For all $T$, $\theta \in \Theta$, and for all $V$ and $V'$ that differ in a single entry, $|q(F(T, \theta, \alpha, R), V) - q(F(T, \theta, \alpha, R), V')| \leq \frac{\beta_2}{m}$.*

Condition (1), the training stability condition, bounds the change in the validation score $q$, when one person's private data in the training set $T$ changes, and the validation set $V$ as well as *the value of the random variable $R$ remains the same*. Our validation procedure critically relies on this condition, and our main contribution in this paper is to identify and exploit it to provide a validation procedure that uses the privacy budget efficiently.

As $F(T, \theta, \alpha, R)$ is a deterministic function, Condition (2), the validation stability condition, bounds the change in $q$ when one person's private data in the validation set $V$ changes, and the output of the training procedure remains the same. We observe that (some version of) Condition (2) is a standard requirement in existing differentially private algorithms that preserve the privacy of the validation dataset while selecting a $h \in \mathcal{C}$ that approximately maximizes $q(h, V)$, even if it is not required to maintain privacy with respect to the training data.

Several remarks are in order. First, observe that Condition (1) is a property of the *differentially private training algorithm* (in addition to $q$ and the non-private quantity being approximated). Even if all else remains the same, different differentially private approximations to the same non-private quantity will have different values of $\beta_1$.

Second, Condition (1) does not always hold for small $\beta_1$ as an immediate consequence of differential privacy of the training procedure. Differential privacy ensures that the *probability of any outcome* is almost the same when the inputs differ in the value of a single individual; Condition (1) requires that even when *the same randomness is used*, the validation score evaluated on the actual output of the algorithm does not change very much when the inputs differ by a single individual's private value.

In Section 6.1, we present an example of a problem and two $\alpha$-differentially private training algorithms which approximately optimize the same function; the first algorithm is based on exponential mechanism, and the second on a maximum of Laplace random variables mechanism. We show that while both provide $\alpha$-differential privacy guarantees, the first algorithm does not satisfy training stability for $\beta_1 = o(n)$ and small enough $\delta$ while the second one ensures training stability for $\beta_1 = 1$ and $\delta = 0$. In Section 4, we present two case studies of commonly used differentially private algorithms where Conditions (1) and (2) hold for constant $\beta_1$ and $\beta_2$.

When the $(\beta_1, \beta_2, \delta)$-stability condition holds, we can design an end-to-end differentially private parameter tuning algorithm, which is shown in Algorithm 2. The algorithm first uses a validation procedure to determine which parameter out of the given set $\Theta$ is (approximately) optimal based on the held-out data (see Algorithm 1). In the next step, the training data is re-used along with the parameter output by Algorithm 1 and fresh randomness to generate the final output. Note that we use $\mathbf{Exp}(\gamma)$ to denote the exponential distribution with expectation $\gamma$.

---
**Algorithm 1** Validate($\Theta$, $\mathcal{T}$, $T$, $V$, $\beta_1$, $\beta_2$, $\alpha_1$, $\alpha_2$)
---
1: **Inputs:** Parameter list $\Theta = \{\theta_1, \ldots, \theta_k\}$, training procedure $\mathcal{T} = (\mathcal{G}, F)$, validation score $q$, training set $T$, validation set $V$, stability parameters $\beta_1$ and $\beta_2$, training privacy parameter $\alpha_1$, validation privacy parameter $\alpha_2$.
2: **for** $i = 1, \ldots, k$ **do**
3:     Draw $R_i \sim \mathcal{G}$. Compute $h_i = F(T, \theta_i, \alpha_1, R_i)$.
4:     Let $\beta = \max(\frac{\beta_1}{n}, \frac{\beta_2}{m})$.
5:     Let $t_i = q(h_i, V) + 2\beta Z_i$, where $Z_i \sim \mathbf{Exp}(\frac{1}{\alpha_2})$.
6: **end for**
7: Output $i^* = \operatorname{argmax}_i t_i$.
---

Algorithm 1 takes as input a training procedure $\mathcal{T}$, a parameter list $\Theta$, a validation score $q$, training and validation datasets $T$ and $V$, and privacy parameters $\alpha_1$ and $\alpha_2$. It runs the training procedure $\mathcal{T}$ on the same training set $T$ with privacy budget $\alpha_1$ for each parameter in $\Theta$ to generate outputs $h_1, h_2, \ldots$, and then uses an $\alpha_2$-differentially private procedure to select the index $i^*$ such that the validation score $q(h_{i^*}, V)$ is (approximately) maximum. For simplicity, we use a *maximum of Exponential random variables* procedure, inspired by [1], to find the approximate maximum; an exponential mechanism [21] may also be used instead. Algorithm 2 then re-uses the training data set $T$ to train with parameter $\theta_{i^*}$ to get the final output.

---
**Algorithm 2** End-to-end Differentially Private Training and Validation Procedure
---
1: **Inputs:** Parameter list $\Theta = \{\theta_1, \ldots, \theta_k\}$, training procedure $\mathcal{T} = (\mathcal{G}, F)$, validation score $q$, training set $T$, validation set $V$, stability parameters $\beta_1$ and $\beta_2$, training privacy parameter $\alpha_1$, validation privacy parameter $\alpha_2$.
2: $i^* = \mathbf{Validate}(\Theta, \mathcal{T}, T, V, \beta_1, \beta_2, \alpha_1, \alpha_2)$.
3: Draw $R \sim \mathcal{G}$. Output $h = F(T, \theta_{i^*}, \alpha_1, R)$.
---

### 3.1 Performance Guarantees

Theorem 1 shows that Algorithm 1 is $(\alpha_2, \delta)$-differentially private, and Theorem 2 shows privacy guarantees on Algorithm 2. Detailed proofs of both theorems are provided in the Supplementary Material. We observe that Conditions (1) and (2) are critical to the proof of Theorem 1.

**Theorem 1 (Privacy Guarantees for Validation Procedure)** *If the validation score $q$ is $(\beta_1, \beta_2, \frac{\delta}{k})$-stable with respect to the training procedure $\mathcal{T}$, the privacy parameter $\alpha_1$ and the parameter set $\Theta$, then, Algorithm 1 guarantees $(\alpha_2, \delta)$-differential privacy.*

**Theorem 2 (End-to-end Privacy Guarantees)** *If the conditions in Theorem 1 hold, and if $\mathcal{T}$ is $\alpha_1$-differentially private, then Algorithm 2 is $(\alpha_1 + \alpha_2, \delta)$-differentially private.*

Theorem 3 shows guarantees on the utility of the validation procedure – that it selects an index $i^*$ which is not too suboptimal.

**Theorem 3 (Utility Guarantees)** *Let $h_1, \ldots, h_k$ be the output of the differentially private training procedure in Step (3) of Algorithm 1. Then, with probability $\geq 1 - \delta_0$, $q(h_{i^*}, V) \geq \max_{1 \leq i \leq k} q(h_i, V) - \frac{2\beta \log(k/\delta_0)}{\alpha_2}$.*

## 4  Case Studies

We next show that Algorithm 2 may be applied to design end-to-end differentially private training and validation procedures for two fundamental statistical and machine-learning tasks – training a linear classifier, and building a histogram density estimator. In each case, we use existing differentially private algorithms and validation scores for these tasks. We show that the validation score satisfies the $(\beta_1, \beta_2, \delta)$-stability property with respect to the training procedure for small values of $\beta_1$ and

$\beta_2$, and thus we can apply in Algorithm 2 with a small value of $\beta$ to obtain end-to-end differential privacy.

Details of the case study for regularized linear classification is shown in Section 4.1, and those for histogram density estimation is presented in the Supplementary Material.

## 4.1 Linear Classification based on Logistic Regression and SVM

Given a set of labelled examples $(x_1, y_1), \ldots, (x_n, y_n)$ where $x_i \in \mathbf{R}^d$, $\|x_i\| \leq 1$ for all $i$, and $y_i \in \{-1, 1\}$, the goal in linear classification is to train a linear classifier that largely separates examples from the two classes. A popular solution in machine learning is to find a classifier $w^*$ by solving a regulared convex optimization problem:

$$w^* = \mathrm{argmin}_{w \in \mathbf{R}^d} \frac{\lambda}{2} \|w\|^2 + \frac{1}{n} \sum_{i=1}^{n} \ell(w, x_i, y_i) \tag{1}$$

Here $\lambda$ is a regularization parameter, and $\ell$ is a convex loss function. When $\ell$ is the logistic loss function $\ell(w, x, y) = \log(1 + e^{-y_i w^\top x_i})$, then we have logistic regression. When $\ell$ is the hinge loss $\ell(w, x, y) = \max(0, 1 - y_i w^\top x_i)$, then we have Support Vector Machines. The optimal value of $\lambda$ is data-dependent, and there is no good pre-defined way to select $\lambda$ apriori. In practice, the optimal $\lambda$ is determined by training a small number of classifiers with different $\lambda$ values, and picking the one that has the best performance on a held-out validation dataset.

[4] present two algorithms for computing differentially private approximations to these regularized convex optimization problems for fixed $\lambda$: output perturbation and objective perturbation. We restate output perturbation as Algorithm 4 (in the Supplementary Material) and objective perturbation as Algorithm 3. It was shown by [4] that provided certain conditions hold on $\ell$ and the data, Algorithm 4 is $\alpha$-differentially private; moreover, with some additional conditions on $\ell$, Algorithm 3 is $\alpha + 2 \log \left(1 + \frac{c}{\lambda n}\right)$-differentially private, where $c$ is a constant that depends on the loss function $\ell$, and $\lambda$ is the regularization parameter.

---

**Algorithm 3** Objective Perturbation for Differentially Private Linear Classification

---

1: **Inputs:** Regularization parameter $\lambda$, training set $T = \{(x_i, y_i), i = 1, \ldots, n\}$, privacy parameter $\alpha$.
2: Let $\mathcal{G}$ be the following density over $\mathbf{R}^d$: $\rho_{\mathcal{G}}(r) \propto e^{-\|r\|}$. Draw $R \sim \mathcal{G}$.
3: Solve the convex optimization problem:

$$w^* = \mathrm{argmin}_{w \in \mathbf{R}^d} \frac{\lambda}{2} \|w\|^2 + \frac{1}{n} \sum_{i=1}^{n} \ell(w, x_i, y_i) + \frac{2}{\alpha n} R^\top w \tag{2}$$

4: Output $w^*$.

---

In the sequel, we use the notation $\mathcal{X}$ to denote the set $\{x \in \mathbf{R}^d : \|x\| \leq 1\}$.

**Definition 3** *A function $g : \mathbf{R}^d \times \mathcal{X} \times \{-1, 1\} \to \mathbf{R}$ is said to be L-Lipschitz if for all $w, w' \in \mathbf{R}^d$, for all $x \in \mathcal{X}$, and for all $y$, $|g(w, x, y) - g(w', x, y)| \leq L \cdot \|w - w'\|$.*

Let $V = \{(\bar{x}_i, \bar{y}_i), i = 1, \ldots, m\}$ be the validation dataset. For our validation score, we choose a function of the form:

$$q(w, V) = -\frac{1}{m} \sum_{i=1}^{m} g(w, \bar{x}_i, \bar{y}_i) \tag{3}$$

where $g$ is an $L$-Lipschitz loss function. In particular, the logistic loss and the hinge loss are 1-Lipschitz, whereas the $0/1$ loss is not $L$-Lipschitz for any $L$. Other examples of 1-Lipschitz but non-convex losses include the ramp loss: $g(w, x, y) = \min(1, \max(0, 1 - yw^\top x))$.

The following theorem shows that any non-negative and $L$-Lipschitz validation score is stable with respect to Algorithms 3 and 4 and a set of regularization parameters $\Lambda$; a detailed proof is provided in the Supplementary Material. Thus we can use Algorithm 2 along with this training procedure

and any $L$-Lipschitz validation score to get an end-to-end differentially private algorithm for linear classification.

**Theorem 4 (Stability of differentially private linear classifiers)** *Let* $\Lambda = \{\lambda_1, \ldots, \lambda_k\}$ *be a set of regularization parameters, let* $\lambda_{\min} = \min_{i=1}^{k} \lambda_i$, *and let* $g^* = \max_{(x,y) \in \mathcal{X}, w \in \mathbf{R}^d} g(w, x, y)$. *If* $\ell$ *is convex and* 1*-Lipschitz, and if* $g$ *is* $L$*-Lipschitz and non-negative, then, the validation score* $q$ *in Equation 3 is* $(\beta_1, \beta_2, \frac{\delta}{k})$*-stable with respect to Algorithms 3 and 4,* $\alpha$ *and* $\Lambda$ *for:*

$$\beta_1 = \frac{2L}{\lambda_{\min}}, \quad \beta_2 = \min\left(g^*, \frac{L}{\lambda_{\min}}\left(1 + \frac{d\log(dk/\delta)}{\alpha n}\right)\right)$$

**Example.** For example, if $g$ is chosen to be the hinge loss, then $\beta_1 = \frac{2}{\lambda_{\min}}$ and $\beta_2 = \frac{1}{\lambda_{\min}}\left(1 + \frac{d\log(dk/\delta)}{\alpha n}\right)$. This follows from the fact that the hinge loss is 1-Lipschitz, but may be unbounded for $w$ of unbounded norm.

If $g$ is chosen to be the ramp loss, then $\beta_1 = \frac{2}{\lambda_{\min}}$, and $\beta_2 = 1$ (assuming that $\lambda_{\min} \le 1$). This follows from the fact that the ramp loss is 1-Lipschitz, but bounded at 1 for any $w$ and $(x, y) \in \mathcal{X}$.

## 5 Experiments

In order to evaluate Algorithm 2 empirically, we compare the regularizer parameter values and performance of regularized logistic regression classifiers the algorithm produces with those produced by four alternative methods. We used datasets from two domains, and used 10 times 10-fold cross-validation (CV) to reduce variability in the computed performance averages.

**The Methods** Each method takes input $(\alpha, \Theta, T, V)$, where $\alpha$ denotes the allowed differential privacy, $T$ is a training set, $V$ is a validation set, and $\Theta = \{\theta_1, \ldots, \theta_k\}$ a list of $k$ regularizer values. Also, let $oplr(\alpha, \lambda, T)$ denote the application of the objective perturbation training procedure given in Algorithm 3 such that it yields $\alpha$-differential privacy.

The first of the five methods we compare is *Stability*, the application of Algorithm 2 with $oplr$ used for learning classifiers, $\delta$ chosen in an ad-hoc manner to be 0.01, average negative ramp loss used as validation score $q$, and with $\alpha_1 = \alpha_2 = \alpha/2$.

The four other methods work by performing the following 4 steps: (1) for each $\theta_i \in \Theta$, train a differentially private classifier $f_i = oplr(\alpha_i, \theta_i, T_i)$, (2) determine the number of errors $e_i$ each $f_i$ makes on validation set $V$, (3) randomly choose $i^*$ from $\{1, 2, \ldots, k\}$ with probability $P(i^* = i|p_i)$, and (4) output $(\theta_{i^*}, f_{i^*})$.

What differentiates the four alternative methods is how $\alpha_i$, $T_i$, and $p_i$ are determined. For *alphaSplit*: $\alpha_i = \alpha/k$, $T_i = T$, $p_i \propto e^{-\alpha e_i/2}$, *dataSplit*: $\alpha_i = \alpha$, partition $T$ into $k$ equally sized sets $T_i$, $p_i \propto e^{-\alpha e_i/2}$ (used in [4]), *Random*: $\alpha_i = \alpha$, $T_i = T$, $p_i \propto 1$, and *Control*: $\alpha_i = \alpha$, $T_i = T$, $p_i \propto \mathbf{1}(i = \arg\max_j q(f_j, V))$. Note that for *alphaSplit*, $\alpha/k > \alpha'$ where $\alpha'$ is the solution of $\alpha = k(e^{\alpha'} - 1)\alpha' + \sqrt{2k\log(1/\delta)}\alpha'$ for all of our experimental settings, except when $\alpha = 0.3$, then $\alpha/k > \alpha' - 0.0003$. The method *Control* is not private, and serves to provide an approximate upper bound on the performance of *Stability*. The three other alternative methods are differentially private which we state in the following theorem.

**Theorem 5 (Privacy of alternative methods)** *If* $T$ *and* $V$ *are disjoint, both alphaSplit and dataSplit are* $\alpha$*-differentially private. Random is* $\alpha$ *differentially private even if* $T$ *and* $V$ *are not disjoint, in which case alphaSplit and dataSplit are* $2\alpha$*-differentially private.*

**Procedures and Data** We performed 10 10-fold CV as follows. For round $i$ in each of the CV experiments, fold $i$ was used as a test set $W$ on which the produced classifiers were evaluated, fold $(i \mod 10) + 1$ was used as $V$, and the remaining 8 folds were used as $T$. Furthermore $k = 10$ with $\Theta = \{0.001, 0.112, 0.223, 0.334, 0.445, 0.556, 0.667, 0.778, 0.889, 1\}$. Note that the order of $\Theta$ is chosen such that $i < j$ implies $\theta_i < \theta_j$. By Theorems 2 and 5, all methods except *Control* produce

a $(\alpha, \delta)$-differentially private classifier. Classifier performance was evaluated using the area under the receiver operator curve [25] (AUC) as well as mean squared error (MSE). All computations were done using the R environment [22], and data sets were scaled such that covariate vectors were constrained to the unit ball. We used the following data available from the UCI Machine Learning Repository [9]:

*Adult* – 98 predictors (14 original including categorical variables that needed to be recoded). The data set describes measurements on cases taken from the 1994 Census data base. The classification is whether or not a person has an annual income exceeding 50000 USD, which has a prevalence of 0.22. Each experiment involves computing more than 24000 classifiers. In order to reduce computation time, we selected 52 predictors using the `step` procedure for a model computed by `glm` with family `binomial` and logit link function.

*Magic* – 10 predictors on 19020 cases. The data set describes simulated high energy gamma particles registered by a ground-based atmospheric Cherenkov gamma telescope. The classification is whether particles are primary gammas (signal) or from hadronic showers initiated by cosmic rays in the upper atmosphere (background). The prevalence of primary gammas is 0.35.

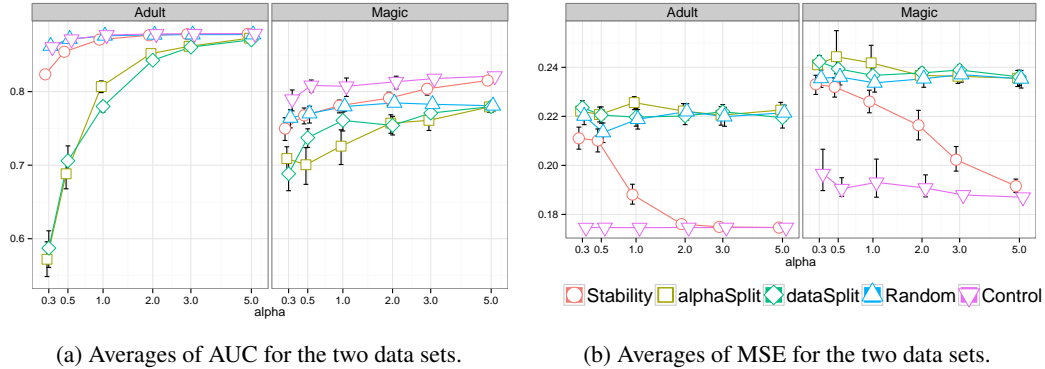

(a) Averages of AUC for the two data sets.  (b) Averages of MSE for the two data sets.

Figure 1: A summary of 10 times 10-fold cross-validation experiments for different privacy levels $\alpha$. Each point in the figure represents a summary of 100 data points. The error bars indiciate a boot-strap sample estimate of the 95% confidence interval of the mean. A small amount of jitter was added to positions on the x-axes to avoid over-plotting.

**Results** Figure 1 summarizes classifier performances and regularizer choices for the different values of the privacy parameter $\alpha$, aggregated over all cross-validation runs. Figure 1a shows average performance in terms of AUC, and Figure 1b shows average performance in terms of MSE.

Looking at AUC in our experiments, *Stability* significantly outperformed *alphaSplit* and *dataSplit*. However, *Stability* only outperformed *Random* for $\alpha > 1$ in the *Magic* data set, and was in fact outperformed by *Random* in the *Adult* data set. In the *Adult* data set, regularizer choice did not seem to matter as *Random* performed equally well to *Control*. For MSE on the other hand, *Stability* outperformed the differentially private alternatives in all experiments. We suggest the following intuition regarding these results. The calibration of a logistic regression model instance, i.e., the difference between predicted probabilities and a 0/1 encoding of the corresponding labels, is not captured well by AUC (or 0/1 error rate) as AUC is insensitive to all strictly monotonically increasing transformations of the probabilities. MSE is often used as a measure of probabilistic model calibration and can be decomposed into two terms: reliability (a calibration term), and refinement (a discrimination measure) which is related to the AUC. In the *Adult* data set, the minor change in AUC of *Control* and *Random* for $\alpha > 0.5$, together with the apparent insensitivity of AUC to regularizer value, suggests that any improvement in *Stability* performance can only come from (the observed) improved calibration. Unlike in the *Adult* data set, there is a AUC performance gap between *Control* and *Random* in the *Magic* data set. This means that regularizer choice matters for discrimination, and we observe improvement for *Stability* in both discrimination and calibration.

**Acknowledgements** This work was supported by NIH grants R01 LM07273 and U54 HL108460, the Hellman Foundation, and NSF IIS 1253942.

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

# 6 Appendix

## 6.1 An Example to Show Training Stability is not a Direct Consequence of Differential Privacy

We now present an example to illustrate that training stability is a property of the training algorithm and not a direct consequence of differential privacy. We present a problem and two $\alpha$-differentially private training algorithms which approximately optimize the same function; the first algorithm is based on exponential mechanism, and the second on a maximum of Laplace random variables mechanism. We show that while both provide $\alpha$-differential privacy guarantees, the first algorithm does not satisfy training stability while the second one does.

Let $i \in \{1, \ldots, l\}$, and let $f : \mathcal{X}^n \times \mathbf{R} \to [0, 1]$ be a function such that for all $i$ and all datasets $D$ and $D'$ of size $n$ that differ in the value of a single individual, $|f(D, i) - f(D', i)| \leq \frac{1}{n}$.

Consider the following training and validation problem. Given a sensitive dataset $D$, the private training procedure $A$ outputs a tuple $(i^*, t_1, \ldots, t_l)$, where $i^*$ is the output of the $\alpha/2$-differentially private exponential mechanism [21] run to approximately maximize $f(D, i)$, and each $t_i$ is equal to $f(D, i)$ plus an independent Laplace random variable with standard deviation $\frac{2l}{\alpha n}$. For any validation dataset $V$, the validation score $q((i^*, t_1, \ldots, t_l), V) = t_{i^*}$.

It follows from standard results that $A$ is $\alpha$-differentially private. Moreover, $A$ can be represented by a tuple $\mathcal{T}_A = (\mathcal{G}_A, F_A)$, where $\mathcal{G}_A$ is the following density over sequences of real numbers of length $l + 1$:

$$\mathcal{G}_A(r_0, r_1, \ldots, r_l) = \mathbf{1}_{0 \leq r_0 \leq 1} \cdot \frac{1}{2^l} e^{-(|r_1| + |r_2| + \ldots + |r_l|)}$$

Thus $\mathcal{G}_A$ is the product of the uniform density on $[0, 1]$ and $l$ standard Laplace densities. Consider the following map $E_0$. For $r \in [0, 1]$, let

$$E_0(r) = i, \quad \text{if } \frac{\sum_{j<i} e^{n\alpha f(D,j)/4}}{\sum_j e^{n\alpha f(D,j)/4}} \leq r \leq \frac{\sum_{j\leq i} e^{n\alpha f(D,j)/4}}{\sum_j e^{n\alpha f(D,j)/4}}$$

In other words, $E_0(r)$ is the map that converts a random number $r$ drawn from the uniform distribution on $[0, 1]$ to the $\alpha/2$-differentially private exponential mechanism distribution that approximately maximizes $f(D, i)$. Given a $l + 1$-tuple $R = (R_0, R_1, \ldots, R_l)$, $F_A$ is now the following map:

$$F_A(D, \alpha, R) = \left( E(R_0), f(D, 1) + \frac{2lR_1}{\alpha n}, f(D, 2) + \frac{2lR_2}{\alpha n}, \ldots, f(D, l) + \frac{2lR_l}{\alpha n} \right)$$

Let $l = 2$ and $D$ and $D'$ be two datasets that differ in the value of a single individual. Suppose it is the case that $f(D, 1) = 1$, $f(D, 2) = \frac{1}{2}$ and $f(D', 1) = 1 - \frac{1}{n}$, $f(D', 2) = \frac{1}{2} + \frac{1}{n}$. Observe that for $D$, the exponential mechanism picks 1 with probability $\frac{e^{n\alpha/4}}{e^{n\alpha/4} + e^{n\alpha/8}}$, and 2 with probability $\frac{e^{n\alpha/8}}{e^{n\alpha/4} + e^{n\alpha/8}}$, where as for $D'$, it picks 1 with probability $\frac{e^{(n-1)\alpha/4}}{e^{(n-1)\alpha/4} + e^{(n+2)\alpha/8}}$ and 2 with probability $\frac{e^{(n+2)\alpha/8}}{e^{(n-1)\alpha/4} + e^{(n+2)\alpha/8}}$. Thus, if $R_0$ lies in the interval $\left[ \frac{e^{(n-1)\alpha/4}}{e^{(n-1)\alpha/4} + e^{(n+2)\alpha/8}}, \frac{e^{n\alpha/4}}{e^{n\alpha/4} + e^{n\alpha/8}} \right]$, then, $F_A(D, \alpha, R) = t_1$ whereas $F_A(D', \alpha, R) = t_2$. When $n$ is large enough, with high probability, $|t_1 - t_2| \geq \frac{1}{3}$; thus, the training stability condition does not hold for $A$ for $\beta_1 = o(n)$ and $\delta < \frac{e^{n\alpha/8}(e^{\alpha/2} - 1)}{(e^{n\alpha/8} + 1)(e^{n\alpha/8} + e^{\alpha/2})}$.

Consider a different algorithm $A'$ which computes $t_1, \ldots, t_l$ first, and then outputs the index $i^*$ that maximizes $t_{i^*}$. Then $A'$ can be represented by a tuple $\mathcal{T}_{A'} = (\mathcal{G}_{A'}, F_{A'})$, where $G_{A'}$ is a density over sequences of real numbers of length $l$ as follows:

$$\mathcal{G}_A(r_1, \ldots, r_l) = \frac{1}{2^l} e^{-(|r_1| + \ldots + |r_l|)}$$

and $F_{A'}$ is the map:

$$F_{A'}(D, \alpha, R) = \left( \text{argmax}_i (f(D, i) + \frac{lR_i}{\alpha n}), f(D, 1) + \frac{lR_1}{\alpha n}, f(D, 2) + \frac{lR_2}{\alpha n}, \ldots, f(D, l) + \frac{lR_l}{\alpha n} \right)$$

For the same value of $R_1, \ldots, R_l$, if $i^* = i$ on input dataset $D$ and if $i^* = i'$ on input dataset $D'$, then, $|f(D, i) - f(D, i')| \le \frac{1}{n}$; this implies that

$$|q(F_{A'}(D, \alpha, R), V) - q(F_{A'}(D', \alpha, R), V)| = |t_i - t_{i'}| = |f(D, i) - f(D', i')| \le \frac{1}{n}$$

with probability 1 over $\mathcal{G}_{A'}$. Thus the training stability condition holds for $\beta_1 = 1$ and $\delta = 0$.

## 6.2 Output Perturbation Algorithm

We present the output perturbation algorithm for regularized linear classification.

---

**Algorithm 4** Output Perturbation for Differentially Private Linear Classification

---

1: **Inputs:** Regularization parameter $\lambda$, training set $T = \{(x_i, y_i), i = 1, \ldots, n\}$, privacy parameter $\alpha$.
2: Let $\mathcal{G}$ be the following density over $\mathbf{R}^d$: $\rho_{\mathcal{G}}(r) \propto e^{-\|r\|}$. Draw $R \sim \mathcal{G}$.
3: Solve the convex optimization problem:

$$w^* = \operatorname{argmin}_{w \in \mathbf{R}^d} \frac{1}{2}\lambda\|w\|^2 + \frac{1}{n}\sum_{i=1}^{n} \ell(w, x_i, y_i) \quad (4)$$

4: Output $w^* + \frac{2}{\lambda \alpha n} R$.

---

## 6.3 Case Study: Histogram Density Estimation

Our second case study is developing an end-to-end differentially private solution for histogram-based density estimation. In density estimation, we are given $n$ samples $x_1, \ldots, x_n$ drawn from an unknown density $f$, and our goal is to build an approximation $\hat{f}$ to $f$. In a histogram density estimator, we divide the range of the data into equal-sized bins of width $h$; if $n_i$ out of $n$ of the input samples lie in bin $i$, then $\hat{f}$ is the density function: $\hat{f}(x) = \sum_{i=1}^{1/h} \frac{n_i}{hn} \cdot \mathbf{1}(x \in \text{Bin } i)$.

A critical parameter while constructing the histogram density estimator is the bin size $h$. There is much theoretical literature on how to choose $h$ – see [16, 26] for surveys. However, the choice of $h$ is usually data-dependent, and in practice, the optimal $h$ is often determined by building a histogram density estimator for a few different values of $h$, and selecting the one which has the best performance on held-out validation data.

The most popular measure to evaluate the quality of a density estimator is the $L_2$-distance or the Integrated Square Error (ISE) between the density estimate and the true density:

$$\|\hat{f} - f\|_2 = \int_x (\hat{f}(x) - f(x))^2 \mathrm{d}x = \int_x f^2(x)\mathrm{d}x + \int_x \hat{f}^2(x)\mathrm{d}x - 2\int_x f(x)\hat{f}(x)\mathrm{d}x \quad (5)$$

$f$ is typically unknown, so the ISE cannot be computed exactly. Fortunately it is still possible to *compare* multiple density estimates based on this distance. The first term in the right hand side of Equation 5 depends only on $f$, and is equal for all $\hat{f}$. The second term is a function of $\hat{f}$ only and can thus be computed. The third term is $2\mathbb{E}_{x \sim f}[\hat{f}(x)]$, and even though it cannot be computed exactly without knowledge of $f$, we can estimate it based on a held out validation dataset. Thus, given a density estimator $\hat{f}$ and a validation dataset $V = \{z_1, \ldots, z_m\}$, we will use the following function to evaluate the quality of $\hat{f}$ on $V$:

$$q(\hat{f}, V) = -\int_x \hat{f}^2(x)\mathrm{d}x + \frac{2}{m}\sum_{i=1}^{m} \hat{f}(z_i) \quad (6)$$

A higher value of $q$ indicates a smaller distance $\|\hat{f} - f\|^2$, and thus a higher quality density estimate. For other measures, see [6].

In the sequel, we assume that the data lies in the interval $[0, 1]$ and that this interval is known in advance. For ease of notation, we also assume without loss of generality that $\frac{1}{h}$ is an integer. For

ease of exposition, we confine ourselves to one-dimensional data, although the general techniques can be easily extended to higher dimensions. Given $n$ samples and a bin size $h$, several works, including [7, 19, 27, 28, 20, 29, 14] have shown different ways of constructing and sampling from differentially private histograms. The most basic approach is to construct a non-private histogram and then add Laplace noise to each cell, followed by some post-processing. Algorithm 5 presents a variant of a differentially private histogram density estimator due to [19] in our framework.

---

**Algorithm 5** Differentially Private Histogram Density Estimator

---

1: **Inputs:** Bin size $h$ (such that $1/h$ is an integer), data $T = \{x_1, \ldots, x_n\}$, privacy parameter $\alpha$.
2: **for** $i = 1, \ldots, \frac{1}{h}$ **do**
3:     Draw $R_i$ independently from the standard Laplace density: $\rho_{\mathcal{G}}(r) = \frac{1}{2}e^{-|r|}$.
4:     Let $I_i = \left[\frac{i-1}{h}, \frac{i}{h}\right)$. Define: $n_i = \sum_{j=1}^{n} \mathbf{1}(x_j \in I_i)$, and let $\tilde{n}_i = \max\left(0, n_i + \frac{2R_i}{\alpha}\right)$.
5: **end for**
6: Let $\tilde{n} = \sum_i \tilde{n}_i$. Return the density estimator: $\hat{f}(x) = \sum_{i=1}^{1/h} \frac{\tilde{n}_i}{h\tilde{n}} \cdot \mathbf{1}(x \in I_i)$

---

The following theorem shows stability guarantees on the differentially private histogram density estimator described in Algorithm 5.

**Theorem 6 (Stability of Private Histogram Density Estimator)** *Let $H = \{h_1, \ldots, h_k\}$ be a set of bin sizes, and let $h_{\min} = \min_i h_i$. For any fixed $\delta$, if the sample size $n \geq 1 + \frac{2\ln(4k/\delta)}{\alpha\sqrt{h_{\min}}}$, then, the validation score $q$ in Equation 6 is $(\beta_1, \beta_2, \frac{\delta}{k})$-Stable with respect to Algorithm 5 and $H$ for: $\beta_1 = \frac{6}{(1-\nu)h_{\min}}$, $\quad \beta_2 = \frac{2}{h_{\min}}$, where: $\nu = \frac{2\ln(4k/\delta)}{n\alpha\sqrt{h_{\min}}}$.*

## 6.4 Proofs of Theorems 1, 2 and 3

We now present the proofs of Theorems 1, 2 and 3. Our proofs involve ideas similar to those in the analysis of the multiplicative weights update method for answering a set of linear queries in a differentially private manner [13].

Let $\mathcal{A}(D)$ denote the output of Algorithm 1 when the input is a sensitive dataset $D = (T, V)$, where $T$ is the training part and $V$ is the validation part. Let $D' = (T', V)$ where $T$ and $T'$ differ in the value of a single individual, and let $D'' = (T, V')$ where $V$ and $V'$ differ in the value of a single individual. The proof of Theorem 1 is a consequence of the following two lemmas.

**Lemma 1** *Suppose that the conditions in Theorem 1 hold. Then, for all $D = (T, V)$, all $D' = (T', V)$, such that $T$ and $T'$ differ in the value of a single individual, and for any set of outcomes $S$:*
$$\Pr(\mathcal{A}(D) \in S) \leq e^{\alpha_2} \Pr(\mathcal{A}(D') \in S) + \delta \tag{7}$$

**Lemma 2** *Suppose that the conditions in Theorem 1 hold. Then, for all $D = (T, V)$, all $D'' = (T, V')$ such that $V$ and $V'$ differ in the value of a single individual, and for any set of outcomes $S$,*
$$\Pr(\mathcal{A}(D) \in S) \leq e^{\alpha_2} \Pr(\mathcal{A}(D'') \in S) + \delta \tag{8}$$

PROOF: (Of Lemma 1) Let $S = (I, C)$, where $I \subseteq [k]$ is a set of indices and $C \subseteq \mathcal{C}$. Let $E$ be the event that all of $R_1, \ldots, R_k$ lie in the set $\Sigma$. We will first show that conditioned on $E$, for all $i$, it holds that:
$$\Pr(i^* = i|D, E) \leq e^{\alpha_2} \Pr(i^* = i|D', E) \tag{9}$$
Since $\Pr(E) \geq 1 - \delta$, from the conditions in Theorem 1, for any subset $I$ of indices, we can write:
$$
\begin{aligned}
\Pr(i^* \in I|D) &\leq \Pr(i^* \in I|D, E)\Pr(E) + (1 - \Pr(E)) \\
&\leq e^{\alpha_2} \Pr(i^* \in I|D', E)\Pr(E) + \delta \\
&\leq e^{\alpha_2} \Pr(i^* \in I, E|D') + \delta \\
&\leq e^{\alpha_2} \Pr(i^* \in I|D') + \delta
\end{aligned} \tag{10}
$$

We will now prove Equation 9. For this purpose, we adopt the following notation. We use the notation $Z_{\backslash i}$ to denote the random variables $Z_1, \ldots, Z_{i-1}, Z_{i+1}, \ldots, Z_k$ and $z_{\backslash i}$ to denote the set of values $z_1, \ldots, z_{i-1}, z_{i+1}, \ldots, z_k$. We also use the notation $h(\cdot)$ to represent the density induced on the random variables $Z_1, \ldots, Z_k$ by Algorithm 1. In addition, we use the notation $R$ to denote the vector $(R_1, \ldots, R_k)$. We first fix a value $z_{\backslash i}$ for $Z_{\backslash i}$, and a value of $R$ such that $R_1, \ldots, R_k$ all lie in $\Sigma$, and consider the ratio of probabilities:

$$\frac{\Pr(i^* = i | Z_{\backslash i} = z_{\backslash i}, D, R)}{\Pr(i^* = i | Z_{\backslash i} = z_{\backslash i}, D', R)}$$

Observe that this ratio of probabilities is equal to:

$$\frac{\Pr(Z_i + q(F(T, \theta_i, \alpha_1, R_i), V) \geq \sup_{j \neq i} z_j + q(F(T, \theta_j, \alpha_1, R_j), V))}{\Pr(Z_i + q(F(T', \theta_i, \alpha_1, R_i), V) \geq \sup_{j \neq i} z_j + q(F(T', \theta_j, \alpha_1, R_j), V))}$$

which is in turn equal to:

$$\frac{\Pr(Z_i \geq \sup_{j \neq i} z_j + q(F(T, \theta_j, \alpha_1, R_j), V) - q(F(T, \theta_i, \alpha_1, R_i), V))}{\Pr(Z_i \geq \sup_{j \neq i} z_j + q(F(T', \theta_j, \alpha_1, R_j), V) - q(F(T', \theta_i, \alpha_1, R_i), V))}$$

Observe that from the stability condition,

$$\begin{aligned}
&|(q(F(T, \theta_j, \alpha_1, R_j), V) - q(F(T, \theta_i, \alpha_1, R_i), V)) - (q(F(T', \theta_j, \alpha_1, R_j), V) - q(F(T', \theta_i, \alpha_1, R_i), V))| \\
\leq\ & |q(F(T, \theta_j, \alpha_1, R_j), V) - q(F(T', \theta_j, \alpha_1, R_j), V')| + |q(F(T, \theta_i, \alpha_1, R_i), V) - q(F(T', \theta_i, \alpha_1, R_i), V)| \\
\leq\ & \frac{2\beta_1}{n} \leq 2\beta
\end{aligned}$$

Thus, the ratio of the probabilities is at most the ratio $\Pr(Z_i \geq \gamma) / \Pr(Z_i \geq \gamma + 2\beta)$ where $\gamma = \sup_{j \neq i} z_j + q(F(T, \theta_j, \alpha_1, R_j), V) - q(F(T, \theta_i, \alpha_1, R_i), V)$, which is at most $e^{\alpha_2}$ by properties of the exponential distribution. Thus, we have established that for all $z_{\backslash i}$, for all $R$ in $\Sigma^k$,

$$\Pr(i^* = i | Z_{\backslash i} = z_{\backslash i}, D, R) \leq e^{\alpha_2} \cdot \Pr(i^* = i | Z_{\backslash i} = z_{\backslash i}, D', R)$$

Equation 9 follows by integrating over $z_{\backslash i}$ and $R$. The lemma follows. $\square$

PROOF:(Of Lemma 2) Let $S = (I, C)$, where $I \subseteq [k]$ is a set of indices and $C \subseteq \mathcal{C}$. Let $E$ be the event that all of $R_1, \ldots, R_k$ lie in $\Sigma$. We will first show that conditioned on $E$, for all $i$, it holds that:

$$\Pr(i^* = i | D, E) \leq e^{\alpha_2} \Pr(i^* = i | D'', E) \tag{11}$$

Since $\Pr(E) \geq 1 - \delta$, from the conditions in Theorem 1, for any subset $I$ of indices, we can write:

$$\begin{aligned}
\Pr(i^* \in I | D) &\leq \Pr(i^* \in I | D, E) \Pr(E) + (1 - \Pr(E)) \\
&\leq e^{\alpha_2} \Pr(i^* \in I | D'', E) \Pr(E) + \delta \\
&\leq e^{\alpha_2} \Pr(i^* \in I, E | D'') + \delta \\
&\leq e^{\alpha_2} \Pr(i^* \in I | D'') + \delta \tag{12}
\end{aligned}$$

We will now focus on showing Equation 11. We first consider the case when event $E$ holds, that is, $R_j \in R$, for $j = 1, \ldots, k$. In this case, the stability definition and the conditions of the theorem imply that for all $\theta_j \in \Theta$,

$$|q(F(T, \theta_j, \alpha_1, R_j), V) - q(F(T, \theta_j, \alpha_1, R_j), V')| \leq \frac{\beta_2}{m} \leq \beta \tag{13}$$

In what follows, we use the notation $Z_{\backslash i}$ to denote the random variables $Z_1, \ldots, Z_{i-1}, Z_{i+1}, \ldots, Z_k$ and $z_{\backslash i}$ to denote the set of values $z_1, \ldots, z_{i-1}, z_{i+1}, \ldots, z_k$. We also use the notation $h(\cdot)$ to represent the density induced on the random variables $Z_1, \ldots, Z_k$ by Algorithm 1. In addition, we use the notation $R$ to denote the vector $(R_1, \ldots, R_k)$. We first fix a value $z_{\backslash i}$ for $Z_{\backslash i}$, and a value of $R$ such that $E$ holds, and consider the ratio of probabilities:

$$\frac{\Pr(i^* = i | Z_{\backslash i} = z_{\backslash i}, D, R)}{\Pr(i^* = i | Z_{\backslash i} = z_{\backslash i}, D'', R)}$$

Observe that this ratio of probabilities is equal to:

$$\frac{\Pr(Z_i + q(F(T, \theta_i, \alpha_1, R_i), V) \geq \sup_{j \neq i} z_j + q(F(T, \theta_j, \alpha_1, R_j), V))}{\Pr(Z_i + q(F(T, \theta_i, \alpha_1, R_i), V') \geq \sup_{j \neq i} z_j + q(F(T, \theta_j, \alpha_1, R_j), V'))}$$

which is in turn equal to:

$$\frac{\Pr(Z_i \geq \sup_{j \neq i} z_j + q(F(T, \theta_j, \alpha_1, R_j), V) - q(F(T, \theta_i, \alpha_1, R_i), V))}{\Pr(Z_i \geq \sup_{j \neq i} z_j + q(F(T, \theta_j, \alpha_1, R_j), V') - q(F(T, \theta_i, \alpha_1, R_i), V'))}$$

Observe that from Equation 13,

$$|(q(F(T, \theta_j, \alpha_1, R_j), V) - q(F(T, \theta_i, \alpha_1, R_i), V)) - (q(F(T, \theta_j, \alpha_1, R_j), V') - q(F(T, \theta_i, \alpha_1, R_i), V'))| \leq \frac{2\beta_2}{m} \leq 2\beta$$

Thus, the ratio of the probabilities is at most the ratio $\Pr(Z_i \geq \gamma)/\Pr(Z_i \geq \gamma + 2\beta)$ for $\gamma = \sup_{j \neq i} z_j + q(F(T, \theta_j, \alpha_1, r_j), V) - q(F(T, \theta_i, \alpha_1, r_i), V)$, which is at most $e^{\alpha_2}$ by properties of the exponential distribution. Thus, we have established that when $R \in \Sigma^k$, for all $j$,

$$\frac{\Pr(i^* = i | Z_{\backslash i} = z_{\backslash i}, D, R)}{\Pr(i^* = i | Z_{\backslash i} = z_{\backslash i}, D'', R)} \leq e^{\alpha_2}$$

Thus for any such $R$, we can write:

$$\frac{\Pr(i^* = i | D, R)}{\Pr(i^* = i | D'', R)} = \frac{\int_{z_{\backslash i}} \Pr(i^* = i | Z_{\backslash i} = z_{\backslash i}, D, R) h(z_{\backslash i}) dz_{\backslash i}}{\int_{z_{\backslash i}} \Pr(i^* = i | Z_{\backslash i} = z_{\backslash i}, D'', R) h(z_{\backslash i}) dz_{\backslash i}} \leq e^{\alpha_2}$$

Equation 11 now follows by integrating $R$ over $E$. $\square$

PROOF:(Of Theorem 1) The proof of Theorem 1 follows from a combination of Lemmas 1 and 2. $\square$

PROOF:(Of Theorem 2) The proof of Theorem 2 follows from privacy composition; Theorem 1 ensures that Step (2) of Algorithm 2 is $(\alpha_2, \delta)$-differentially private; moreover the training procedure $\mathcal{T}$ is $\alpha_1$-differentially private. The theorem follows by composing these two results. $\square$

PROOF:(Of Theorem 3) Observe that:

$$\Pr\left(q(h_{i^*}, V) < \max_{1 \leq i \leq k} q(h_i, V) - \frac{2\beta \log(k/\delta_0)}{\alpha_2}\right) \leq \Pr\left(\exists j \text{ s.t. } Z_j \geq \frac{\log(k/\delta_0)}{\alpha_2}\right)$$

By properties of the exponential distribution, for any fixed $j$, $\Pr(Z_j \geq \frac{\log(k/\delta_0)}{\alpha_2}) \leq \frac{\delta_0}{k}$. Thus the theorem follows by an Union Bound. $\square$

## 6.5 Proof of Theorem 4

PROOF: (Of Theorem 4 for Output Perturbation) Let $T$ and $T'$ be two training sets which differ in a single labelled example $((x_n, y_n)$ vs. $(x'_n, y'_n))$, and let $w^*(T)$ and $w^*(T')$ be the solutions to the regularized convex optimization problem in Equation 1 when the inputs are $T$ and $T'$ respectively. We observe that for fixed $\lambda$, $\alpha$ and $R$,

$$F(T, \lambda, \alpha, R) - F(T', \lambda, \alpha, R) = w^*(T) - w^*(T')$$

When the training sets are $T$ and $T'$, the objective functions in the regularized convex optimization problems are both $\lambda$-strongly convex, and they differ by $\frac{1}{n}(\ell(w, x_n, y_n) - \ell(w, x'_n, y'_n))$. Combining this fact with Lemma 1 of [4], and using the fact that $\ell$ is 1-Lipschitz, we have that for all $\lambda$ and $R$,

$$\|F(T, \lambda, \alpha, R) - F(T', \lambda, \alpha, R)\| \leq \frac{2}{\lambda n}$$

Since $g$ is $L$-Lipschitz, this implies that for any fixed validation set $V$, and for all $\lambda$, $\alpha$ and $R$,

$$|q(F(T, \lambda, \alpha, R), V) - q(F(T', \lambda, \alpha, R), V)| \leq \frac{2L}{\lambda n} \tag{14}$$

Now let $V$ and $V'$ be two validation sets that differ in the value of a single labelled example $(\bar{x}_m, \bar{y}_m)$. Since $g \geq 0$ for all inputs, for any such $V$ and $V'$, and for a fixed $\Lambda$, $\alpha$ and $R$, $|q(F(T, \lambda, \alpha, R), V) - q(F(T, \lambda, \alpha, R), V')| \leq \frac{g_{\max}}{m}$, where

$$g_{\max} = \sup_{(x,y) \in \mathcal{X}} g(F(T, \lambda, \alpha, R), x, y)$$

By definition, $g_{\max} \leq g^*$. Moreover, as $g$ is $L$-Lipschitz,

$$g_{\max} \leq L \cdot \|F(T, \lambda, \alpha, R)\|$$

Now, let $E$ be the event that $\|R\| \leq d \log(dk/\delta)$. From Lemma 4 of [4], $\Pr(E) \geq 1 - \delta/k$. Thus, provided $E$ holds, we have that:

$$\|F(T, \lambda, \alpha, R)\| \leq \|w^*\| + \frac{d \log(dk/\delta)}{\lambda \alpha n} \leq \frac{1}{\lambda} + \frac{d \log(dk/\delta)}{\lambda \alpha n} = \frac{1}{\lambda} \left(1 + \frac{d \log(dk/\delta)}{n\alpha}\right)$$

where the bound on $\|w^*\|$ follows from an application of Lemma 1 of [4] on the functions $\frac{1}{2}\lambda\|w\|^2$ and $\frac{1}{2}\lambda\|w\|^2 + \frac{1}{n}\sum_{i=1}^{n} \ell(w, x_i, y_i)$. This implies that provided $E$ holds, for all training sets $T$, and for all $\lambda$,

$$|q(F(T, \lambda, \alpha, R), V) - q(F(T, \lambda, \alpha, R), V')| \leq \frac{L}{\lambda m} \left(1 + \frac{d \log(dk/\delta)}{n\alpha}\right) \qquad (15)$$

The theorem now follows from a combination of Equations 14 and 15, and the definition of $g^*$. $\square$

PROOF: (Of Theorem 4 for Objective Perturbation) Let $T$ and $T'$ be two training sets which differ in a single labelled example $(x_n, y_n)$. We observe that for a fixed $R$ and $\lambda$, the objective of the regularized convex optimization problem in Equation 2 differs in the term $\frac{1}{n}(\ell(w, x_n, y_n) - \ell(w, x'_n, y'_n))$. Combining this with Lemma 1 of [4], and using the fact that $\ell$ is 1-Lipschitz, we have that for all $\lambda$, $\alpha$, $R$,

$$\|F(T, \lambda, \alpha, R) - F(T', \lambda, \alpha, R)\| \leq \frac{2}{\lambda n}$$

Since $g$ is $L$-Lipschitz, this implies that for any fixed validation set $V$, and for all $\lambda$ and $r$,

$$|q(F(T, \lambda, \alpha, R), V) - q(F(T', \lambda, \alpha, R), V)| \leq \frac{2L}{\lambda n} \qquad (16)$$

Now let $V$ and $V'$ be two validation sets that differ in the value of a single labelled example $(\bar{x}_m, \bar{y}_m)$. Since $g \geq 0$, for any such $V$ and $V'$, $|q(F(T, \lambda, \alpha, R), V) - q(F(T, \lambda, \alpha, R), V')| \leq \frac{g_{\max}}{m}$, where

$$g_{\max} = \sup_{(x,y) \in \mathcal{X}} g(F(T, \lambda, \alpha, R), x, y)$$

By definition $g_{\max} \leq g^*$. Moreover, as $g$ is $L$-Lipschitz,

$$g_{\max} \leq L \cdot \|F(T, \lambda, \alpha, R)\|$$

Let $E$ be the event that $\|R\| \leq d \log(dk/\delta)$. From Lemma 4 of [4], $\Pr(E) \geq 1 - \delta/k$. Thus, provided $E$ holds, we have that:

$$\|F(T, \lambda, \alpha, R)\| \leq \frac{1 + \|R\|/(\alpha n)}{\lambda} \leq \frac{1}{\lambda} \left(1 + \frac{d \log(dk/\delta)}{n\alpha}\right)$$

This implies that provided $E$ holds, for all training sets $T$, and for all $\lambda$,

$$|q(F(T, \lambda, \alpha, R), V) - q(F(T, \lambda, \alpha, R), V')| \leq \frac{L}{\lambda m} \left(1 + \frac{d \log(dk/\delta)}{n\alpha}\right) \qquad (17)$$

The theorem now follows from a combination of Equations 16 and 17, and the definition of $g^*$. $\square$

## 6.6 Proof of Theorem 6

**Lemma 3 (Concentration of Sum of Laplace Random Variables)** *Let $Z_1, \ldots, Z_s$ be $s \geq 2$ iid standard Laplace random variables, and let $Z = Z_1 + \ldots + Z_s$. Then, for any $\theta$,*

$$\Pr(Z \geq \theta) \leq \left(1 - \frac{1}{s}\right)^{-s} e^{-\theta/\sqrt{s}} \leq 4e^{-\theta/\sqrt{s}}$$

PROOF: The proof follows from using the method of generating functions. The generating function for the standard Laplace distribution is: $\psi(X) = \mathbb{E}[e^{tX}] = \frac{1}{1-t^2}$, for $|t| \leq 1$. As $Z_1, \ldots, Z_s$ are independently distributed, the generating function for $Z$ is $\mathbb{E}[e^{tZ}] = (1-t^2)^{-s}$. Now, we can write:

$$\begin{aligned}
\Pr(Z \geq \theta) &= \Pr(e^{tZ} \geq e^{t\theta}) \\
&\leq \frac{\mathbb{E}[e^{tZ}]}{e^{t\theta}} = e^{-t\theta} \cdot (1-t^2)^{-s}
\end{aligned}$$

Plugging in $t = \frac{1}{\sqrt{s}}$, we get that:

$$\Pr(Z \geq \theta) \leq \left(1 - \frac{1}{s}\right)^{-s} e^{-\theta/\sqrt{s}}$$

The lemma follows by observing that for $s \geq 2$, $(1 - \frac{1}{s})^s \geq \frac{1}{4}$. $\square$

PROOF: (Of Theorem 6) Let $V = \{z_1, \ldots, z_m\}$ be a validation dataset, and let $V'$ be a validation dataset that differs from $V$ in a single sample ($z_m$ vs $z'_m$). We use the notation $R$ to denote the sequence of values $R = (R_1, R_2, \ldots, R_{1/h})$. Given an input sample $T$, a bin size $h$, a privacy parameter $\alpha$, and a sequence $R$, we use the notation $\hat{f}_{T,h,\alpha,R}$ to denote the density estimator $F(T, h, \alpha, R)$. For all such $T$, all $h$, all $\alpha$ and all $R$, we can write:

$$\begin{aligned}
|q(F(T, h, \alpha, R), V) - q(F(T, h, \alpha, R), V')| &= \frac{2}{m}(\hat{f}_{T,h,\alpha,R}(z_m) - \hat{f}_{T,h,\alpha,R}(z'_m)) \\
&\leq \frac{2}{m} \cdot \frac{\max_i \tilde{n}_i}{h\tilde{n}} \leq \frac{2}{mh}
\end{aligned} \tag{18}$$

For a fixed value of $h$, we define the following event $E$:

$$\sum_{i=1}^{1/h} R_i \geq -\frac{\ln(4k/\delta)}{\sqrt{h}}$$

Using the symmetry of Laplace random variables and Lemma 3, we get that $\Pr(E) \geq 1 - \delta/k$. We observe that provided the event $E$ holds,

$$\tilde{n} \geq n - \sum_{i=1}^{1/h} R_i \geq n - \frac{2\ln(4k/\delta)}{\alpha\sqrt{h}} \geq n(1 - \nu) \tag{19}$$

Let $T$ and $T'$ be two input datasets that differ in a single sample ($x_n$ vs $x'_n$). We fix a bin size $h$, a value of $\alpha$, and a sequence $R$, and for these fixed values, we use the notation $\tilde{n}_i$ and $\tilde{n}'_i$ to denote the value of $\tilde{n}_i$ in Algorithm 5 when the inputs are $T$ and $T'$ respectively. Similarly, we use $\tilde{n} = \sum_i \tilde{n}_i$ and $\tilde{n}' = \sum_i \tilde{n}'_i$.

For any $V$, we can write:

$$\begin{aligned}
q(F(T, h, \alpha, R), V) - q(F(T', h, \alpha, R), V) &= \frac{2}{m} \sum_{j=1}^{m} (\hat{f}_{T,h,\alpha,R}(z_j) - \hat{f}_{T',h,\alpha,R}(z_j)) \\
&\quad - \sum_{i=1}^{1/h} h \cdot \left(\frac{\tilde{n}_i^2}{h^2\tilde{n}^2} - \frac{\tilde{n}_i'^2}{h^2\tilde{n}'^2}\right)
\end{aligned} \tag{20}$$

We now look at bounding the right hand side of Equation 20 term by term. Suppose $T'$ is obtained rom $T$ by moving a single sample $x_n$ from bin $a$ to bin $b$ in the histogram. Then, depending on the relative values of $\tilde{n}_a$ and $\tilde{n}_b$, there are four cases:

1. $\tilde{n}_a' = \tilde{n}_a - 1$, $\tilde{n}_b' = \tilde{n}_b + 1$. Thus $\tilde{n}' = \tilde{n}$.
2. $\tilde{n}_a' = \tilde{n}_a = 0$, $\tilde{n}_b' = \tilde{n}_b + 1$. Thus $\tilde{n}' = \tilde{n} + 1$.
3. $\tilde{n}_a' = \tilde{n}_a - 1$, $\tilde{n}_b' = \tilde{n}_b = 0$. Thus $\tilde{n}' = \tilde{n} - 1$.
4. $\tilde{n}_a' = \tilde{n}_a = 0$, $\tilde{n}_b' = \tilde{n}_b = 0$. Thus $\tilde{n}' = \tilde{n}$.

In the fourth case, $\hat{f}_{T,h,\alpha,R} = \hat{f}_{T',h,\alpha,R}$, and thus the right hand side of Equation 20 is 0. Moreover, the second and the third cases are symmetric. We thus focus on the first two cases.

In the first case, the first term in the right hand side of Equation 20 can be written as:

$$\left| \frac{2}{m} \cdot \sum_{j=1}^{m} \sum_{i=1}^{1/h} \mathbf{1}(z_j \in I_i) \cdot \left( \frac{\tilde{n}_i}{h\tilde{n}} - \frac{\tilde{n}_i'}{h\tilde{n}'} \right) \right| = \left| \frac{2}{m} \cdot \sum_{j=1}^{m} \sum_{i=1}^{1/h} \mathbf{1}(z_j \in I_i) \cdot \frac{\tilde{n}_i - \tilde{n}_i'}{h\tilde{n}} \right|$$

$$\leq \frac{2}{m} \cdot m \cdot \frac{1}{h\tilde{n}} \leq \frac{2}{h\tilde{n}}$$

The second term on the right hand side of Equation 20 can be written as:

$$\left| \sum_{i=1}^{1/h} \left( \frac{\tilde{n}_i^2}{h\tilde{n}^2} - \frac{\tilde{n}_i'^2}{h\tilde{n}'^2} \right) \right| = \frac{\tilde{n}_a^2 + \tilde{n}_b^2 - (\tilde{n}_a - 1)^2 - (\tilde{n}_b + 1)^2}{h\tilde{n}^2}$$

$$= \left| \frac{2\tilde{n}_a - 2\tilde{n}_b - 2}{h\tilde{n}^2} \right| \leq \frac{2}{h\tilde{n}}$$

where the last step follows from the fact that $\tilde{n}_b' = \tilde{n}_b + 1 \leq \tilde{n}$. Thus, for the first case, the right hand side of Equation 20 is at most $\frac{4}{h\tilde{n}}$.

We now consider the second case. The first term on the right hand side of Equation 20 can be written as:

$$\left| \frac{2}{m} \cdot \sum_{j=1}^{m} \sum_{i=1}^{1/h} \mathbf{1}(z_j \in I_i) \cdot \left( \frac{\tilde{n}_i}{h\tilde{n}} - \frac{\tilde{n}_i'}{h\tilde{n}'} \right) \right|$$

$$= \left| \frac{2}{mh} \cdot \sum_{j=1}^{m} \sum_{i=1}^{1/h} \mathbf{1}(z_j \in I_i) \cdot \left( \frac{\tilde{n}_i}{\tilde{n}} - \frac{\tilde{n}_i'}{\tilde{n} + 1} \right) \right|$$

$$\leq \frac{2}{hm} \cdot m \cdot \frac{1}{\tilde{n}(\tilde{n} + 1)} \cdot \max(|\tilde{n}_i(\tilde{n} + 1) - \tilde{n}_i\tilde{n}|, |\tilde{n}_i(\tilde{n} + 1) - \tilde{n}(\tilde{n}_i + 1)|)$$

$$\leq \frac{2}{h} \cdot \frac{1}{\tilde{n}(\tilde{n} + 1)} \cdot \max(|\tilde{n}_i|, |\tilde{n} - \tilde{n}_i|) \leq \frac{2}{h(\tilde{n} + 1)}$$

where the last step follows from the fact that $\max(|\tilde{n}_i|, |\tilde{n} - \tilde{n}_i|) \leq \tilde{n}$. The second term on the right hand side of Equation 20 can be written as:

$$\left| \sum_{i=1}^{1/h} \left( \frac{\tilde{n}_i^2}{h\tilde{n}^2} - \frac{\tilde{n}_i'^2}{h\tilde{n}'^2} \right) \right| = \sum_{i \neq b} \left( \frac{\tilde{n}_i^2}{h\tilde{n}^2} - \frac{\tilde{n}_i^2}{h(\tilde{n} + 1)^2} \right) + \left| \frac{\tilde{n}_b^2}{h\tilde{n}^2} - \frac{(\tilde{n}_b + 1)^2}{h(\tilde{n} + 1)^2} \right|$$

$$= \frac{2\tilde{n} + 1}{h\tilde{n}^2(\tilde{n} + 1)^2} \cdot \sum_{i \neq b} \tilde{n}_i^2 + \left| \frac{(\tilde{n}_b - \tilde{n})(2\tilde{n}_b\tilde{n} + \tilde{n} + \tilde{n}_b)}{h\tilde{n}^2(\tilde{n} + 1)^2} \right|$$

$$\leq \frac{2\tilde{n} + 1}{h(\tilde{n} + 1)^2} + \frac{\tilde{n} \cdot 2\tilde{n}(\tilde{n} + 1)}{h\tilde{n}^2(\tilde{n} + 1)^2} \leq \frac{4}{h(\tilde{n} + 1)}$$

Thus, in the second case, the right hand side of Equation 20 is at most $\frac{6}{h(\tilde{n}+1)}$. We observe that the third case is symmetric to the second case, and thus we can carry out very similar calculations in the third case to show that the right hand side is at most $\frac{6}{h\tilde{n}}$. Thus, we have that for any $T$ and $T'$, provided the event $E$ holds,

$$|q(F(T, h, \alpha, R), V) - q(F(T', h, \alpha, R), V)| \leq \frac{6}{h\tilde{n}} \tag{21}$$

The theorem now follows by combining Equation 21 with Equation 19. $\square$

## 6.7 Proof of Theorem 5

**Lemma 4** *(Parallel construction) Let $\mathbb{A} = \{\mathcal{A}_1, \mathcal{A}_2, \ldots, \mathcal{A}_k\}$ be a list of $k$ independently randomized functions, and let $\mathcal{A}_i$ be $\alpha_i$-differentially private. Let $\{D_1, D_2, \ldots, D_k\}$ be $k$ subsets of a set $D$ such that $i \neq j \implies D_i \cap D_j = \emptyset$. Algorithm $\mathcal{B}(D, \mathbb{A}) = (\mathcal{A}_1(D_1), \mathcal{A}_2(D_2), \ldots, \mathcal{A}_k(D_k))$ is $\max_{1 \leq i \leq k} \alpha_i$-differentially private.*

PROOF: Let $D, D'$ be two datasets such that their symmetric difference contains one element. We have that

$$\frac{P(\mathcal{B}(D, \mathbb{A}) \in S)}{P(\mathcal{B}(D', \mathbb{A}) \in S)} = \frac{P(\mathcal{B}(D, \mathbb{A}) \in S_1 \times \cdots \times S_k)}{P(\mathcal{B}(D', \mathbb{A}) \in S_1 \times \cdots \times S_k)} = \frac{P(\mathcal{A}_1(D_1) \in S_1) \cdots P(\mathcal{A}_k(D_k) \in S_k)}{P(\mathcal{A}_1(D_1') \in S_1) \cdots P(\mathcal{A}_k(D_k') \in S_k)}$$
(22)

by independence of randomness in the $\mathcal{A}_i$. Since $i \neq j \implies D_i \cap D_j = \emptyset$, there exists at most one index $j$ such that $D_j \neq D_j'$. If $j$ does not exist, (22) reduces to $e^0 \leq e^{\max_{1 \leq i \leq k} \alpha_i}$. Let $j$ exist, then

$$\frac{P(\mathcal{B}(D, \mathbb{A}) \in S)}{P(\mathcal{B}(D', \mathbb{A}) \in S)} = \frac{P(\mathcal{A}_j(D_j) \in S_j)}{P(\mathcal{A}_j(D_j') \in S_j)} \leq e^{\alpha_j} \leq e^{\max_{1 \leq i \leq k} \alpha_i},$$

which concludes the proof. $\square$

PROOF: (Theorem 5) We begin by separating task (a) of producing the $f_i$ in step 1. from the task (b) of computing $e_i$ in step 2. and selecting $i^*$ in step 3.

From the parallel construction Lemma 4 it follows that (a) in $dataSplit$ is $\alpha$-differentially private. From standard composition of privacy it follows that (a) in $alphaSplit$ is $\alpha$-differentially private.

Task (b) is for both $alphaSplit$ and $dataSplit$ an application of the exponential mechanism [21], which for choosing with a probability proportional to $\epsilon(-e_i)$ yields $2\epsilon\Delta$-differential privacy, where $\Delta$ is the sensitivity of $e_i$. Since a single change in $V$ can change the number of errors any fixed classifier can make by at most $1 = \Delta$, we get that task (b) is $\alpha$-differentially private for $\epsilon = \alpha/2$.

If $T$ and $V$ are disjoint, we get by parallel construction that both $alphaSplit$ and $dataSplit$ yield $\alpha$-differential privacy. If $T$ and $V$ are not disjoint, by standard composition of privacy we get that both $alphaSplit$ and $dataSplit$ yield $2\alpha$-differential privacy.

In $Random$, the results of step 2. in task (b) are never used in step 3. Step 3 is done without looking at the input data and does not incur loss of differential privacy. We can therefore simulate $Random$ by first choosing $i^*$ uniformly at random, and then computing $f_i$ at $\alpha$-differential privacy, which by standard privacy composition is $\alpha$-differentially private. $\square$

## 6.8 Experimental selection of regularizer index

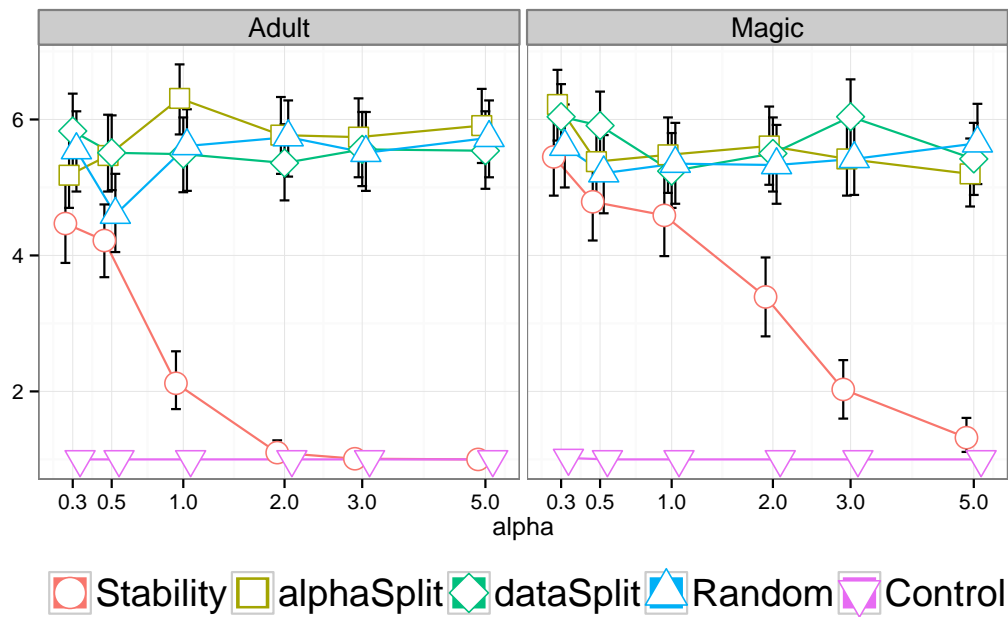

Figure 2: A summary of 10 times 10-fold cross-validation selection of regularizer index $i$ into $\Theta$ for different privacy levels $\alpha$. Each point in the figure represents a summary of 100 data points. The error bars indiciate a boot-strap sample estimate of the 95% confidence interval of the mean. A small amount of jitter was added to positions on the x-axes to avoid over-plotting.

