[Reviews · NeurIPS 2013]

Submitted by Assigned_Reviewer_1

This paper provides a generic way to perform parameter tuning of a given training algorithm in a deferentially private manner.
Given a set of examples (for training and validation), a set of model parameters, a training algorithm, and a performance measure, the proposed procedure outputs a deferentially private hypothesis with respect to the prescribed privacy parameter.
The basis behind the procedure is the definition of (\beta_1, \beta_2, \delta)-stability; which describes the stability of the performance with respect to change in the training set and the validation set.
The procedure basically follows the exponential mechanism, and the utility bound is also provided.

The idea is nice and validation procedures for deferentially private predictor that does not waste privacy budget is an important research direction. I think this is the first well-considered validation procedures for deferentially private predictors.

The paper is well organized but sometimes hard to follow mainly due to confusing notations.

The proposed method was evaluated with only two datasets.
From Fig 1 (a), in the Adult data set, the AUC of the proposed method is not different form that of RANDOM.
In the Magic data set, the proposed method is slightly better than the other methods if \epsilon is larger than 2.0, but such a large epsilon is not practical. I think authors need to evaluate the proposed method with various datasets.
Summary: The idea behind the paper is interesting and promising whereas the superiority of the proposed method to existing methods was not confirmed well from the experimental results.

Submitted by Assigned_Reviewer_6

A Stability-Based Validation Procedure for Differentially Private Machine Learning
------------------------------------------------------------------------------------------------------
The paper discusses the problem of finding in a differentially private manner which hypothesis out of a given set of k hypotheses (often represented as k potential values for a parametrized learning problem, like SVM) gives a better score on a separate validation set. The overall approach is straight-forward: you try all k hypothesis and publish the one that achieves minimal error perturbed by Laplace noise (or highest perturbed score). Clearly, such approach is differentially private only if one has the guarantees that a single person cannot have a significant effect on the score of the best hypothesis (or the hypothesis itself), a premise which the authors refer to as stability. The authors then apply their paradigm to the specific cases of linear classifiers and histogram estimators. Finally, the authors evaluate their technique empirically comparing it to various other differentially private estimators (like naively trying all k assumptions with \alpha=\alpha/k, or splitting the data into k random chunks training each hypothesis on a different chunk).

The paper address an important problem and gives a nice and clean solution to it. The authors take the time to compare their technique to a non-private version of the algorithm, and the paper is really well written. Also, the general problem statement is nice. If we want to learn the best of k hypotheses while preserving \alpha-DP, there are 2 strawmen algorithms: to learn each while preserving (\alpha/k)-DP, which leads to poly(k)-degradation in performance, or to learn each in a manner that preserves \alpha-DP on a 1/k fraction of the data, which causes the sample bounds to grow by a factor of k. The authors' observation is that DP machine learners often output an hypothesis that only slightly changes w.r.t a change of a single individual, so the validation score of each of the k-hypothesis isn't prone to change by much and adding a small amount of noise to the validation score of each hypothesis is therefore enough to preserve DP.

The two major disadvantages of the paper, in my opinion, lie in the fact that the observation is cute but not very surprising -- the stability is a by-product of the fact that the learning algorithms are DP. Because the hypotheses outputted by the learning on any two neighboring training datasets are close, then their validation score is close too. The second problem is in the experiment: for small \alpha (the more interesting regime) is seems that this approach does almost as well as picking a random hypothesis. (Also, why doesn't hypothesis "control" has \alpha=0?) I would much rather see a different experiment, in which the data is synthetically generated so that some hypothesis best explains it, and the authors check how many examples are required in each technique until each of the \alpha-DP algorithms reached error<\epsilon. If indeed the number of samples improves dramatically with the new technique, that would serve as a demonstration that this approach is best.

Also, I object to the use of the term stability. Since q is a function, I do not understand what's wrong with the more concrete notion of Lipfshitzness or smoothness.

Still, all in all it is an interesting paper. I recommend weak acceptance.
Summary: The paper discusses the problem of finding in a differentially private manner which hypothesis out of a given set of k hypotheses gives a better score on a separate validation set. The approach is straight-forward and its based on the observation that previous DP algorithms for ML output similar hypotheses for neighboring datasets whose score don't change significantly.

Submitted by Assigned_Reviewer_7

Differential privacy is a concept that tries to protect sensitive information by
limiting the influence of each data point on the result. This makes it difficult
to infer the data point provided by a single individual, although aggregate
results remain quite accurate and useful. The authors propose a new
differentially private algorithm for tuning the parameters of a classifier. Its
advantage compared to other approaches is that the privacy budget is only split
in two parts for training and validation, but not into many small portions for
each possible choice of the parameters. Therefore the noise added in order to
protect the privacy can be weaker, which improves the accuracy of the tuned
classifier. In the paper, this effect is demonstrated using two real data sets.

The paper is well written and describes the proposed algorithm in detail. It
also contains enough information about differential privacy, so that even
readers unfamiliar with that concept can understand it. However, the
privacy-preserving property of the proposed algorithm is only claimed in the main
paper, but not shown there. Instead the proofs can be found in the supplementary
material.

Section 4.2 introduces a second topic into this paper: estimation of histograms
in a differentially private way. In my opinion, it would be better to leave that
section out, as there are no results shown in the paper. Instead, one could
present the proof (or at least a short version) for theorems 1 and 2 there.

Although the paper only demonstrates one application of the algorithm for
differentially private parameter tuning (Algorithm 2), it can be combined with
other learning algorithms, if a suitable validation score is available.
Therefore it could help to do machine learning on sensitive data sets in general,
while respecting the privacy of individuals.

In their response the authors propose to replace figure 1(b) with a
plot of the average squared error. This change improves the main paper,
as it makes the advantage of the algorithm more visible.
Summary: Good paper with detailed information on the proposed algorithm. High impact as
differentially private parameter tuning can be applied to other machine learning
tasks besides logistic regression and SVM classification.
Author Feedback

Author rebuttal: We thank all the reviewers for their thoughtful comments and feedback.

In the review by Assigned_Reviewer_6, there appears to be a misunderstanding about the main result; perhaps we can clarify. Differential privacy requires that given two adjacent databases D and D', the *probabilities* that a private algorithm produces the same output on D and D' is close. This does not automatically imply the kind of stability that we require in the training procedure, namely, that for all random values taken by the randomization procedure, the *outputs themselves* are close, *conditioned on the _same_ value of the randomization*.

In particular, we can design an explicit example using the exponential mechanism of McSherry and Talwar, where (our notion of) stability is **not** a by-product of differential privacy. However, our notion of stability is indeed a by-product of **certain classes** of differentially private algorithms, such as those used in regularized LR and histogram construction. We feel that further work is needed to characterize when stability is a by-product of differential privacy.

Our key contribution in this paper is to formalize this stability requirement, and show that it allows better validation by efficient use of the privacy budget. We will add the example mentioned above, and make sure to clarify this point better in the final version.

In the experimental results, the measured AUCs were not strongly
dependent on the choice of the regularizer value. This was
particularly evident in the "Adult" data, where a random choice of
value performed equally well as the
non-private control. Also, the AUC scores for the control
and random improved little for values of alpha > 0.5 in both datasets,
suggesting that the logistic regression models were performing close to
optimal with respect to discrimination as measured by AUC.

In addition to AUC, we also computed the mean squared error for the
experiments. The mean squared error is often used as a measure of
probabilistic model calibration and can be decomposed into two terms:
reliability (a calibration term), and refinement (a discrimination
measure) which is related to the AUC. Our method outperformed all other differentially private methods when considering the mean squared error. We present this result in the text and show a plot in the supplement.

In summary, the experimental results suggest that our method outperformed the differentially private alternatives, but since there was little room for improvement in discrimination, this was only evident in model calibration and shown in the mean squared error.

We will replace the Figure 1 (b) by the mean squared error plot
currently found in Figure 2 in the supplement, and move Figure 1 (b)
to the supplement.